# Characterization of the spontaneous degenerative mitral valve disease in FVB mice

**Estelle Ayme-Dietrich**[1], **Sylvia Da Silva**[1], **Ghina Alame Bouabout**[2], **Alizée Arnoux**[1], **Jérôme Guyonnet**[3], **Guillaume Becker**[1], **Laurent Monassier**[1]*

**1** Laboratoire de Pharmacologie et Toxicologie NeuroCardiovasculaire UR7296, Fédération de Médecine Translationnelle, Centre Hospitalier Universitaire et Université de Strasbourg, Centre de Recherche en Biomédecine de Strasbourg (CRBS), Strasbourg, France, **2** Mouse Clinical Institute, Illkirch Cedex, France, **3** Pharma Innovation Department, CEVA Santé Animale, Libourne Cedex, France

* laurent.monassier@unistra.fr

**Data Availability Statement:** All relevant data are within the manuscript and its Supporting Information files.

## Abstract

### Background

The development of new non-surgical treatments dedicated to mitral valve degeneration is limited by the absence of relevant spontaneous and rapidly progressing animal experimental models.

### Animals

We characterized the spontaneous mitral valve degeneration in two inbred FVB mouse strains compared to C57BL/6J and investigated a contribution of the serotonergic system.

### Methods

Males and females FVB/NJ and FVB/NRj were compared to the putative C57BL/6J control at 12, 16, 20 and 24 weeks of age. Body weight, systolic blood pressure, heart rate, urinary 5-hydroxyindoleacetic acid (5-HIAA), whole blood and plasma serotonin, tail bleeding time, blood cell count, plasma TGF-β1 and plasma natriuretic peptide concentrations were measured. Myocardium and mitral valves were characterized by histology. mRNA mitral expression of 5-HT$_{2A}$ and 5-HT$_{2B}$ receptors was measured in the anterior leaflet. Cardiac anatomy and function were assessed by echocardiography.

### Results

Compared to C57BL/6J, FVB mice strains did not significantly differ regarding body weight increase, arterial blood pressure and heart rate. A progressive augmentation of plasma pro-ANP was observed in FVB mice. Nevertheless, no cardiac hypertrophy or left-ventricular fibrosis were observed. Accordingly, plasma TGF-β1 was not different among the three strains. Conversely, FVB mice demonstrated a high prevalence of fibromyxoid highly cellularized and enriched in glycosaminoglycans lesions, inducing major mitral leaflets thickening without increase in length. The increased thickness was correlated with urinary 5-HIAA and blood platelet count. Whole blood serotonin concentration was similar in the two strains but, in FVB, a reduction of plasma serotonin was observed together with an increase of the

**Funding:** Part of this work was financed by CEVA Santé Animale (Libourne, France). There was no additional external funding received for this study. The funder did not provide any support in the form of salaries for authors except JG who is a CEVA employee. CEVA did not have any role in the study design, data collection and analysis, or decision to publish. Only one CEVA member (JG) participated in the preparation of the manuscript.

**Competing interests:** The authors have declared that no competing interests exist.

bleeding time. Finally, echocardiography identified left atrial and left ventricular remodeling associated with thickening of both mitral leaflets and mitral insufficient in 30% of FVB mice but no systolic protrusion of mitral leaflets towards the atrium.

## Conclusion

The FVB mouse strain is highly prone to spontaneous mitral myxomatous degeneration. A contribution of the peripheral serotonergic system is suggested.

## Introduction

Cardiac valve degeneration is now considered as an active phenomenon originating from matrix cells, resident or not, having a potential of differentiation towards myofibroblasts and osteoblasts. *In vivo* experimental modeling of valvular heart disease is both an issue and a difficulty. Human valve degeneration is a usually slow kinetic phenomenon, taking decades to express itself symptomatically. Even if the animal species used for research age faster than humans, a rapid response is often expected, in a timescale in the order of a few weeks. In fact, even in mice, the whole process lasts several months and mixes evolutionary stages of variable kinetics. The safety pharmacology team of the GlaxoSmithKline laboratories addressed this question by following for two years Sprague-Dawley Hsd rats (112 males and 112 females) and Swiss Crl: CD-1 (ICR) BR mice (120 males and 120 females) from the Charles River farm in St-Constent, Quebec (Canada) [1]. Despite the observation of histological lesions, no death could be attributed to a cardiac cause including valvular heart disease. They appeared as fibromyxoid thickening localized towards the distal part of the leaflets. Even if some lesions were observed in the first year, it is important to note that the majority of affected animals were found after the age of 500 days, the frequency of fibromyxoid valve damage being 50% in rats and only 15% in mice with no obvious causal link to cardiomyopathy. As a consequence, most researchers focused on genetically engineered animals showing reproducible and rapidly evolving valve degeneration. Even if of high interest, these animals only model the pathophysiology linked to the gene mutated and not the whole pathophysiological process of spontaneous cardiac valve degeneration. Interestingly, it appeared that in Sprague-Dawley rats, a serotonergic component participates in the development of the lesions since these are enriched in cells expressing 5-HT$_{2B}$ receptors [2, 3]. These data are in accordance with some observations made in the spontaneous mitral valve degeneration of the dog. Small breeds of dogs (Yorkshire, Lhasso Apso) develop degenerative damage to the mitral valve more frequently than larger breeds. This disease reaches a dramatic frequency in the Cavalier King Charles. Classically the valves lose their flexibility, increase their surface and appear bulging, lesions usually qualified as fibromyxoids. Here also, it seems that the pathophysiology of this disease involves activation of the serotonergic system because the serum serotonin concentrations are high in these animals compared to controls [4]. In the valve tissue of diseased animals, an increase in the expression of the serotonergic receptor 5-HT$_{2B}$, in parallel with a repression of its transporter SERT, has also been detected [5]. To find a mouse model of spontaneous mitral valve degeneration with a contribution of the serotonergic system, we screened some inbred mouse strains and found the FVB one. The aim of the present report is to present the main cardiovascular characteristics of this strain with a particular focus on mitral valves and the systemic serotonergic system.

## Material and methods

### Experimental design

In order to characterize a mouse model of spontaneous degenerative mitral valve disease, two FVB mice strains, provided by Charles River from the Jackson's Laboratory (FVB/NJ) and Janvier's laboratories (FVB/NRj) were used and compared to the putative C57BL/6J control. Animals were followed between 12 and 24 weeks of age in parallel groups. For each strain (FVB/NJ, FVB/NRj and C57BL/6J), we built a group of 40 mice (20 for each sex), then divided in 5 males, 5 females, i.e.10 animals per time point, at 12, 16, 20 and 24 weeks of age. At each time-point body weight, systolic blood pressure and heart rate were measured. Urine was collected to dose 5-hydroxyindoleacetic acid (5-HIAA). A terminal blood collection was performed to make a blood cell count and to dose the cytokine TGF-β1 and the natriuretic peptide, pro-ANP. Then, the heart was collected for histological characterization of the myocardium and mitral valves. All animal care and procedures are in accordance with institutional guidelines and European regulations. The protocol has been submitted to French regulation authorities and ethical committees according to the European guidelines. It has been approved by the Strasbourg's ethics committee (CREMEAS) and obtained its authorization from the Ministère de l'Enseignement Supérieur, de la Recherche et de l'Innovation (MESRI) with the following registration number 11732–2017121309379999. In an additional set of animals, mice were anesthetized by 2% isoflurane to perform a bleeding time test. After full recovery, they were sedated by 20mg/kg intraperitoneal xylazine (Rompun©) and then euthanized by a lethal intraperitoneal dose of 200 mg/kg pentobarbital (Euthasol©). Blood was immediately collected by direct cardiac puncture and the anterior leaflet of the mitral valve removed. This complementary protocol has been approved by the ethics committee of the Institute of Genetics and Molecular and Cellular Biology (Cometh) and obtained its authorization from the MESRI with the registration number 26957–2020082617147993.

### Cardiovascular evaluation

Heart rate and blood pressure were recorded by the tail-cuff method (BP2100, Visitech, USA). The mean values of systolic blood pressure (SBP) and heart rate (HR) were obtained at each time-point, after one-week long training.

### Tail bleed assay

The tail bleeding time was obtained in 2% isoflurane anesthetized mice. The tail was rapidly cut 1cm from its tip and dived in a tube containing water at 37˚C. The bleeding time was measured as the delay between the bleeding recovery in water to its full stop.

### Echocardiography and Doppler

To allow rapid recovery from anesthesia, transthoracic echocardiography was performed in mice anesthetized with 1.5–2% isoflurane and placed on a dedicated warming blanket using a Vevo 2100 (FUJIFILM VisualSonics Inc., Toronto, Canada) equipped with the MS-400 30-MHz sectorial transducer. Short- and long-axis views and four chambers apical cardiac views were used for measurements by combining 2D and M-mode imaging. In the long axis view, M-mode imaging allowed the measurements of aorta (AD) and pulmonary artery diameters (PAD). By using pulsed-Doppler imaging we obtained the pulmonary flow and measured its velocity time integral (VTI) and the heart rate (HR). We calculated cardiac output (CO) as $CO = 2\pi \times (PAD/2)^2 \times VTI \times HR$. Cardiac output was also calculated from the aorta diameter and the flow recorded in the ascending aorta from a 2D imaging of the aortic arch. On the

same view, we obtained from the M-mode tracing the thickness and motion of the posterior mitral valve. We measured its thickness 3 times just before its opening. End-diastolic (EDLVD) and end-systolic (ESLVD) left-ventricular diameters, diastolic anterior (AW) and septal (SW) wall thicknesses were obtained from M-mode tracings of the LV in the short-axis view at the level of the papillary muscles. Shortening fraction (SF) was calculated as SF = (EDLVD-ESLVD)/EDLVD. LV weight (LVW) was calculated as LVW = 1.04 x[(EDLVD+AW +SW)$^3$-EDLVD$^3$]. Left-atrial diameter and area were measured from the 4 chambers apical view excluding left auricle in end-diastole. Mitral inflow was recorded by placing the pulsed Doppler window at the tip of the mitral valve. We measured maximal velocities of the E and A waves to obtain E/A ratio and the deceleration time of the E wave. The isovolumetric relaxation time (IVRT) was defined as the interval between aortic closure and the start of mitral inflow. By color Doppler, on the same apical view, we searched the presence of a mitral insufficiency and we obtained from the M-mode tracing the thickness and motion of the anterior mitral valve. Here also, we measured its thickness 3 times just before its opening. All measured and calculated indexes were presented as the average of 3 consecutive beats.

## Urinary 5-HIAA and blood 5-HT dosage

Mice were housed individually in metabolic cages overnight. Urinary 5-HIAA was measured from 0.5–1.0 mL urine collected in a light-protected tube and containing a drop of 5N HCl. Whole blood and plasma 5-HT concentrations were obtained from blood collected by intra-cardiac puncture. Briefly, a direct cardiac puncture was made by the mean of a G26 needle connected to a heparinized syringe in deeply anesthetized mice immediately after respiratory failure. 1mL blood was obtained and separated in two parts: 300μL for whole blood measurements on a citrated tube and 700μL for plasma measurements. The last was centrifuged at 6000 rpm during 10 minutes and the plasma fraction collected and placed in a citrated tube. Global platelet 5-HT content was calculated as whole blood 5-HT minus plasma 5-HT concentrations. Urinary 5-HIAA and blood 5-HT concentrations were obtained by high-performance liquid chromatography (HPLC) (Plateau Technique de Biologie, Nouvel Hôpital Civil, Strasbourg).

## Blood cytokines and pro-ANP measurements

At each time-point and after a lethal pentobarbital intraperitoneal injection (150 mg/kg) (Euthasol®, Le Vet, Netherlands), 0.5 mL of blood was collected by intracardiac puncture. The blood count was measured in whole blood by a clinical blood cell counter (Advia 120, Siemens AG, Erlangen, Germany), with a veterinary software. The plasma cytokines and pro-ANP natriuretic peptide dosages were performed after centrifugation (2000g, 10min) of 0.5ml of total blood. Concentrations of TGF-β1 and pro-ANP were measured in plasma by ELISA kits (Promega, France).

## Mitral valve collection and mRNA quantification by RT-ddPCR

8 mitral valves from C57BL/6J and 10 mitral valves from FVB/NJ were harvested and stored in RNA later (Qiagen, USA) until RNA extraction. Briefly, after euthanasia, the heart was rapidly collected and placed in a solution of RNA later (Qiagen, USA). Then RNA later was injected in the left atrium after puncture and its proper injection in the left ventricle was followed by its ejection through the ascending aorta. The heart was incised longitudinally at the level of the median part of the lateral wall of the left ventricle from the top of the left atrium to the apex. The anterior leaflet of the mitral valve was separated from the septum by gently pulling the appendages that attach the valve to the tip of the anterior papillary muscle. Then the valve was

dissected with the tip of thing scissors at the level of the leaflet insertion on the mitral annulus. Then the tissue was placed in RNA later at -20˚C until use.

Total RNA was extracted as described in Lindner et al. 2021 [6]. For ddPCR, all primers were designed and synthesized as described in Lindner et al., 2021 [7]. *Htr2a* primer sequences are forward: 5'-CCTGAAAATCATTGCGGTGTG-3', reverse: 5'-TGCCACAAAAGAGCC TATGAG-3' and probe 56-FAM/ATCCATGCC/Zen/AATCCCAGTCTTCGG/3IABkFQ. *Htr2b* primer sequences are forward: 5'-TCCTTGGCGATAGCAGATTTG-3', reverse: 5'-TGATG GAGGCAGTTGAAAAGAG-3' and probe /5HEX/TG TGA TGC C/Zen/G ATT GCC CTC TTG A/3IABkFQ. *Hprt* primer and probe sequence are: probe CTTGCTGGTGAAAAGGACC TCTCGAA and primers forwards: 5'-CCCCAAAATGGTTAAGGTTGC-3', reverse: 5'-AA CAAAGTCTGGCCTGTATCC-3'. RNA reverse transcription, droplet generation, PCR amplification, droplets quantification and analysis are described in Lindner *et al* 2020. Results are presented as a ratio of the *Htr2a* or *Htr2b* RNA transcripts and *Hprt* RNA transcripts. Experiments were performed following dMIQE guidelines for reporting ddPCR experiments [8].

## Cardiac mitral valve evaluation by histology

Hearts harvested from mice, were immediately rinsed in saline solution, then weighted and fixed with a 10% formalin solution for 2 days before being processed. For morphometric analysis, 10 sagittal sections (4 μm in thickness) are obtained after paraffin embedding and staining with hematoxylin and eosin (Mice Clinical Institute, Illkirch-Graffenstaden). In each section, mitral valves are examined, measured for length and thickness. Extreme care is taken in sectioning the heart so that the valves are mainly cut longitudinally (with the attachment sides of the leaflets visible on both ends of the valve). Valve histological morphometric analysis was performed visually using a microscope (Leica DM750®, Germany) with a 10x or 40x calibrated objective connected to a camera (Leica DFC 425 C, Germany) and a data acquisition and analysis software (Leica /Microsystems®LAS V4.8), by a single operator (SDaS). Practically, on each section, we applied a grid made of 25 μm$^2$ squares and quantified cells located in each square. For quantification, the whole mitral valve leaflet section is divided in three equal parts. The proximal region is the one nearest the base cusp, followed by the medial and the distal regions. Results are shown as the mean of all 3 segments. The proximal, medial and distal part thicknesses are measured on three distinct sites in each region. All results are expressed in μm. In order to characterize the matrix extracellular composition of the mitral valve lesions, Alcian blue staining was performed on the series slides. In order to assess mitral valve lesions severity, the following scoring analysis was applied: 0 = no mitral valve lesion, 1 = thickness between 50 and 75 μm measured in mitral valve lesion, 2 = thickness between 75 and 125 μm measured in mitral valve lesion, 3 = thickness between 125 et 200 μm, measured in mitral valve lesion and 4 = severe mitral valve lesion greater than 200μm in thickness.

## Cardiac fibrosis evaluation

Interstitial myocardial collagen content was analyzed on picro-Sirius red stained paraffin sections. Cardiac images were captured with the aid of a light camera-equipped microscope, with a fivefold magnification for larger myocardial areas. For each mice's cardiac slice, two images of different focal fibrosis areas and two images outside these areas were recorded for interstitial fibrosis measurements. Interstitial fibrosis detection was performed using a color threshold method under Image J software, marking the red stained collagen fibers and then converting it into a pixel number (Mice Clinical Institute, Illkirch-Graffenstaden). Interstitial fibrosis was then normalized by the total surface (in pixels) of the analyzed region and expressed in percentage. The epicardium, coronary vessels and papillary muscles were removed from images before quantification.

## Statistical analysis

All values are presented as means ± SEM. All statistical analyses were performed with the GraphPad Prism software (version 6.0, La Jolla, CA, USA). Considering that the variance homogeneity was not certain due to the effective equal to or less than 10 mice/group, statistical comparisons between two or more groups were performed using Kruskal-Wallis test (ordinary one-way ANOVA) followed by post-hoc analysis with Dunn's test to compare controls (C57BL/6J) to FVB mice. To assess the correlation between two quantitative values, non-parametric Pearson correlations were performed (not being able to ensure that the data followed a Normal distribution). $P < 0.05$ was considered as statistically significant.

## Results

### Body weight and cardiovascular parameters

The body weight from C57BL/6J and FVB mice was followed in 12-, 16-, 20- and 24- week-old animals. All mice grew for the 12 weeks of the observation period (Table 1) and males were heavier whatever the strain. Regarding the body weight increase and basal values, no difference was observed in a given sex. At the same timepoints, SBP and HR were recorded in conscious mice by tail-cuff photoplethysmography (Table 1). Values were in the usual range of

**Table 1. Physical, hemodynamical and biochemical parameters obtained in males (A) and females (B) of the three strains at the four timepoints.**

**A. Males**

| | W12 | | | W16 | | | W20 | | | W24 | | |
|---|---|---|---|---|---|---|---|---|---|---|---|---|
| | C57BL/6J | FVB/NJ | FVB/NRj | C57BL/6J | FVB/NJ | FVB/NRj | C57BL/6J | FVB/NJ | FVB/NRj | C57BL/6J | FVB/NJ | FVB/NRj |
| BW | 25.4±0.5 | 26.1±1.1 | 27.1±0.5 | 28.9±0.4 | 29.7±0.5 | 29.8±0.1 | 28.4±0.8 | 31.1±0.9 | 30.8±0.9 | 30.8±0.9 | 31.4±1.0 | 31.6±0.4 |
| SBP | 103±7 | 88±5 | 107±11 | 118±5 | 100±7 | 113±4 | 93±6 | 105±4 | 95±6 | 107±4 | 110±5 | 102±4 |
| HR | 641±12 | 737±20 | 66ac2±32 | 662±19 | 735±8 | 697±28 | 688±17 | 713±12 | 686±5 | 715±12 | 701±11 | 641±21 |
| Pro-ANP | 1.2±0.3 | 1.3±0.2 | 0.7±0.2 | 0.3±0.1 | 1.1±0.4 | 0.5±0.1 | 0.5±0.1 | 0.7±0.4 | 1.0±0.3 | 0.4±0.0 | 1.1±0.4 | 0.6±0.1 |
| TGF | 32±8 | 97±20 | 6.2±2.8 | 3.9±0.2 | 29±20 | 4.4±0.3 | 17±12 | 6.8±1.9 | 12±6 | 19±11 | 32±18 | 11±7 |
| HIAA | 66±10 | 69±4 | 57±2 | 42±7 | 61±5 | 73±5 | 46±6 | 80±2 | 71±8 | 32±4 | 48±10 | 40±3 |
| Platelets | 1185±28 | 1357±100 | 943±90 | 1040±50 | 1545±47 | 1432±40 | 847±70 | 1201±90 | 1122±90 | | | |
| HCT | 47±1 | 41±1 | 40±3 | 42±1 | 45±3 | 41±1 | 40±1 | 40±4 | 39±2 | | | |
| HGB | 12.1±0.1 | 11.3±0.3 | 11.2±0.9 | 11.9±0.3 | 12.3±0.6 | 11.8±0.1 | 10.7±0.3 | 11±1 | 11.1±0.5 | | | |

**B. Females**

| | W12 | | | W16 | | | W20 | | | W24 | | |
|---|---|---|---|---|---|---|---|---|---|---|---|---|
| | C57BL/6J | FVB/NJ | FVB/NRj | C57BL/6J | FVB/NJ | FVB/NRj | C57BL/6J | FVB/NJ | FVB/NRj | C57BL/6J | FVB/NJ | FVB/NRj |
| BW | 19.6±0.3 | 20.4±0.7 | 21.6±0.4 | 19.6±0.5 | 24.2±0.9 | 23.9±0.4 | 21.7±0.5 | 25.3±0.4 | 26.5±1.1 | 23.0±0.3 | 26.2±0.9 | 28.8±0.4 |
| SBP | 112±7 | 97±2 | 116±7 | 106±5 | 112±8 | 116±2 | 102±3 | 98±8 | 89±4 | 103±7 | 106±4 | 96±2 |
| HR | 661±14 | 707±8 | 718±13 | 643±38 | 698±12 | 662±20 | 643±29 | 714±5 | 682±32 | 684±9 | 730±16 | 674±16 |
| Pro-ANP | 0.6±0.1 | 0.8±0.2 | 0.7±0.3 | 0.6±0.3 | 0.3±0.1 | 0.6±0.1 | 0.2±0.1 | 0.5±0.2 | 0.5±0.1 | 0.4±0.1 | 1.2±0.4 | 0.9±0.3 |
| TGF | 17±6 | 15±8 | 4.4±0.6 | 16±6 | 5.1±0.4 | 24±21 | 9.7±4.5 | 6.8±1.9 | 4.9±1.3 | 23±10 | 25±6 | 19±12 |
| HIAA | 42±4 | 60±6 | 63±8 | 81±18 | 78±4 | 73±4 | 76±15 | 78±5 | 83±10 | 73±17 | 67±14 | 85±6 |
| Platelets | 776±107 | 1098±85 | 1078±72 | 752±123 | 1125±85 | 1126±53 | 784±51 | 1019±99 | 1038±42 | | | |
| HCT | 42.9±2.4 | 42.9±1.3 | 43.1±1.1 | 38.2±2.2 | 40.0±2.2 | 37.7±1.1 | 45.2±1.6 | 40.7±1.7 | 41.5±1.6 | | | |
| HGB | 11.7±0.6 | 12.2±0.2 | 12.4±0.3 | 10.9±0.1 | 11.5±0.6 | 11.0±0.3 | 12.7±0.5 | 11.3±0.5 | 11.6±0.4 | | | |

BW: body weight (g), SBP: systolic blood pressure (mmHg), HR: heart rate (bpm), Pro-ANP: plasma pro natriuretic peptide type A (nmol/L), TGF: plasma transforming growth factor-β1 ($x10^3$ pg/dL), HIAA: urinary 5-hydroxyindole acetic acid (μmol/L), Platelets ($x10^3$ cells/μL), HCT: hematocrit (%), HGB: hemoglobin (g/dL). Results are presented as means±SEM. No statistically significant difference could be found (Kruskal-Wallis non parametric test, followed by Dunn's multiple comparisons test, p>0.05, n = 5).

normotensive mice and no difference appeared even if males and females of a given strain were pooled. Therefore, no phenotype regarding the blood pressure was observed. Conversely, a tachycardia was regularly observed in FVB as compared to C57BL/6J. This difference reached statistical significance when sexes were pooled at given timepoints: W12: 649±9 bpm in C57BL/6J vs 722±11 bpm in FVB/NJ (p = 0.0006) and 688±20 bpm in FVB/NRj (p = 0.026), W16: 653±20 bpm in C57BL/6J vs 717±9 bpm in FVB/NJ (p = 0.022) but not with FVB/NRj (680±17 bpm; P>0.05) and W20: 667±18 bpm in C57BL/6J vs 713±7 bpm in FVB/NJ (p = 0.042) but not with FVB/NRj (684±15 bpm, p>0.05). No difference was observed at W24 (699±9 bpm in C57BL/6J vs 714±10 bpm in FVB/NJ (p>0.05) and 658±14 bpm in FVB/NRj (p>0.05)). Therefore, tachycardia was more prominent in FVB/NJ than FVB/NRj. To evaluate cardiac stress, we calculated the rate x pressure product that is the product of the SBP and the HR, a parameter correlated to myocardial oxygen consumption. No difference was observed between C57BL/6J and FVB mice meaning that myocardial oxygen demand is similar between the two strains.

## Urinary 5-hydroxyindoleacetic acid measurement (5-HIAA), blood 5-HT and hematological parameters

Taking into account that most of the free plasma serotonin that is metabolized to 5-HIAA is coming from the release of platelet stores, we measured urinary 5-HIAA and blood platelet count in each sex, strain and at W12, W16, W20 and W24. C57BL/6J, FVB/NJ and FVB/NRj exhibited high level of urinary 5-HIAA (**Table 1**) [9]. 5-HIAA concentration reached statistical significance in pooled FVB/NJ vs C57BL/6J at W20 (respectively 79±3 μmol/L and 61±9 μmol/L, p = 0.023). Regarding the blood cell count of white blood cells, neutrophils, eosinophils, lymphocytes, monocytes and red blood cells, we did not observe any difference whatever the age or the sex of the animals. Hemoglobin and hematocrit were also in the same range in all groups (**Table 1**). Conversely, at W16 and 20 the platelet count was significantly increased in FVB males and females compared to C57BL/6J (**Fig 1**). At W24, we faced problems of sampling in FVB mice. Despite this increase of the platelet count, the bleeding time was importantly prolonged in FVB/NJ mice being 139±13 sec compared with C57BL/6J (87±5 sec; p = 0.0002) attesting a possible platelet dysfunction. To move further into the investigation of the peripheral serotonergic system in our animals, we measured whole blood and plasma 5-HT. The whole blood 5-HT concentration was similar in C57BL/6J compared with FVB/NJ

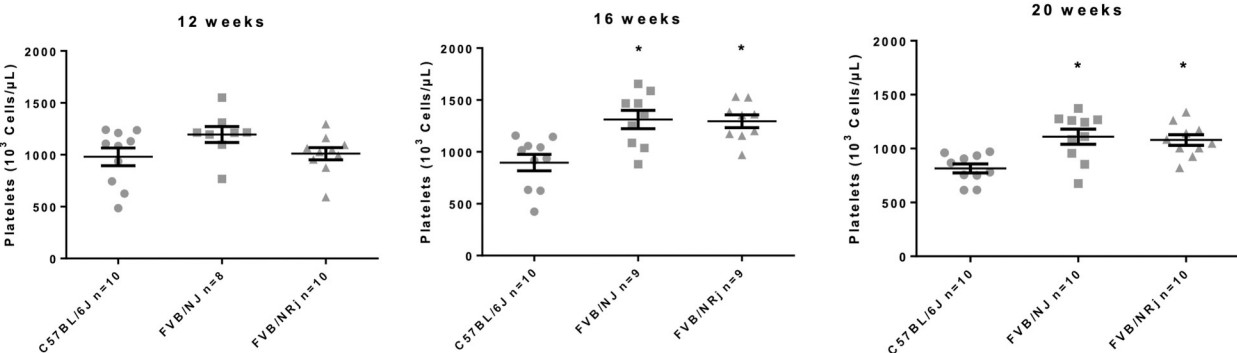

**Fig 1. Platelet count in C57BL/6J and FVB/NJ mice at 12, 16, 20 and 24 weeks.** Platelet count at different times points: W12, W16 and W20, that is significantly increased in pooled FVB compared to C57BL/6J at W16 and W20 (p = 0.0028 and p = 0.0031 respectively). Major blood clotting occurred in FVB mice at W24. Taking into account that platelet count was therefore underestimated due to platelet consumption, results are not presented.

being respectively 3544±348 μmol/L and 3568±196 μmol/L (p>0.05). Conversely, we observed a massive reduction of plasma free 5-HT in FVB/NJ (165±31 μmol/L versus 1172±181 μmol/L in C57BL/6J; p = 0.00007). As a consequence, global platelet serotonin content was increased in FVB/NJ animals (3403±203 μmol/L versus 2372±466 μmol/L in C57BL/6J; p = 0.019).

## Mitral valve expression of 5-HT$_{2A}$ et 5-HT$_{2B}$ receptors

5-HT$_{2A}$ and 5-HT$_{2B}$ receptors are both implied in the fibromyxoid degeneration of the mitral valve. In some species an overexpression of these receptors was previously described. Therefore, we analyzed the expression of these two receptors at the mRNA level in the anterior leaflet of the mitral valve. Both were found in the two strains analyzed. In C57BL/6J the expression of the 5-HT$_{2A}$ receptor is higher than for the 2B one being respectively 0.011±0.001 htr2A cDNA/Hprt cDNA and 0.003±0.001 htr2B cDNA/Hprt cDNA (p = 0.0011). A similar difference was observed in FVB/NJ animals: 0.011±0.001 htr2A cDNA/Hprt cDNA and 0.003 ±0.001 htr2B cDNA/Hprt cDNA (p = 0.0005). Nevertheless, no difference was observed concerning the expression of 5-HT$_{2A}$ and 5-HT$_{2B}$ receptors when comparing C57BL/6J and FVB/ NJ mice.

## Plasma pro-ANP and TGF-β1

Plasma pro-ANP was not different between C57BL/6J and the two FVB substrains at W12 and W16 (**Table 1**). An increase in FVB compared to C57BL/6J reached statistical significance at W24. At that time the pooled male/female concentration was 0.4±0.05 nmol/L in C57BL/6J vs 1.1±0.3 nmol/L in FVB/NJ (p = 0.026) and 0.7±0.1 nmol/L in FVB/NRj (p = 0.032) indicating a possible increase in cardiac filling pressures.

Regarding plasma TGF-β1 (**Table 1**), we observed a very important variability of the results among groups making a group interpretation difficult. Noteworthy, compared to a global mean of 18±4 x10$^3$ pg/dL over the 12 weeks observation period, some animals were showing very high plasma values over 65 x10$^3$ pg/dL only in the FVB groups. Nevertheless, concentrations between 30 and 65 x10$^3$ pg/dL were also found in the C57BL/6J groups.

## Evaluation of myocardial and mitral valve remodeling

To investigate a possible cardiac remodeling, we measured cardiac hypertrophy and myocardial collagen content. At all timepoints the heart weight to body weight ratio was similar among groups without any statistically significant difference, ruling out cardiac hypo or hypertrophy (**S1 Fig**). Then, we evaluated histologically the amount of collagen (i.e. fibrosis) in the left-ventricular myocardium by Sirius red staining. When results were expressed as a percentage of collagen par surface unit, we observed, as a consequence of cardiomyocyte growth, a progressive decrease in all strains between W12 and W20. Nevertheless, no difference was observed between groups at any given time point (**S2 Fig**).

Thickness and length of mitral valve leaflets were analyzed histologically by hematoxylin and eosin staining between 12 and 24 weeks in males and females. At any timepoint, it was never possible to identify differences in length between either FVB substrains and C57BL/6J, neither by comparing the strains nor by comparing the sexes within the same strain (**Fig 2A**). Whatever the measurement time, we were able to identify a significant increase in the thickness of the mitral leaflets in all FVB mice compared to C57BL/6J. Not having observed a difference between males and females, we present the results by pooling the two sexes for each strain (**Fig 2B**). Thus, the mitral leaflets are thickened in FVB/NJ and in FVB/NRj versus C57BL/6J respectively +53.9% and +55.0% at the age of 12 weeks, +44.9% and +66.8% at the age of 16 weeks, +64.0% and +38.2% at the age of 20 weeks and +91.8% and +70.2% at the age

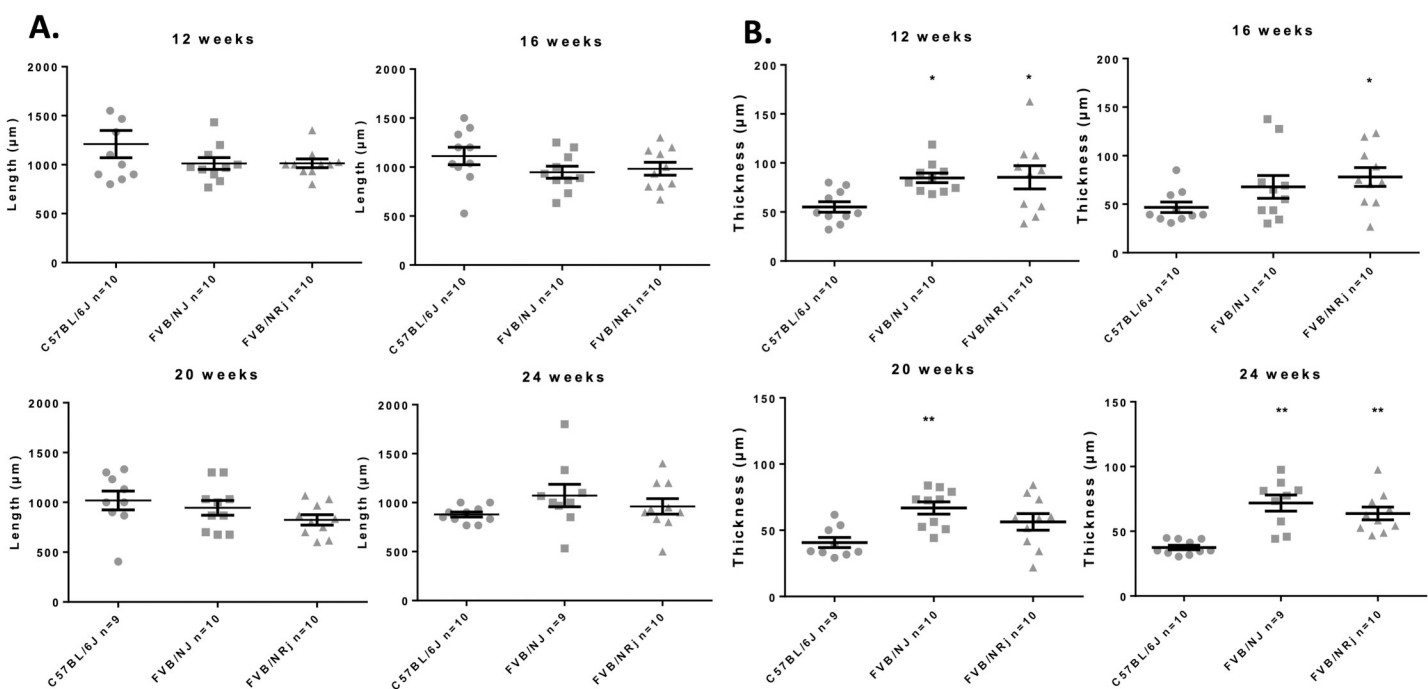

**Fig 2. Characteristics of mitral valve remodeling in FVB mice.** A: Length of the mitral valve leaflets did not differ between FVB mice compared to C57BL/6J mice at any time-point (at 12-, 16-, 20- and 24-weeks) (p>0.05) indicating the absence of mitral valve prolapse in FVB mice. B: Thickness of the mitral valve leaflets in FVB mice substrains increases comparatively to C57BL/6J mice at 12-weeks (p = 0.0131 and p = 0.0375 for respectively FVB/NJ and FVB/NRj vs C57BL/6J), 16-weeks (FVB/NRj vs C57BL/6J, p = 0.0374), 20-weeks (FVB/NJ vs C57BL/6J, p = 0.0052) and 24-weeks (p = 0.0002 and p = 0.0011 for respectively FVB/NJ and FVB/NRj vs C57BL/6J).

of 24 weeks. Unexpectedly, we observed spontaneous mitral valve lesions in our control animals since 30 to 50% of C57BL/6J mice showed lesions (**Fig 3A and 3B**). However, FVB mice are much more affected since 50 to 100% of animals have lesions; at 24 weeks of age, 100% of FVB/NRj mice were showing mitral lesions. Moreover, lesions observed in FVB are also more serious since, if C57BL/6J have severity scores around 1, FVB mice have scores between 2 and 3 (W12: 1.3±0.4 in C57BL/6J vs 2.5±0.2 in FVB/NJ (P>0.05) and 2.3±0.3 in FVB/NRj (p>0.05); W16: 1.1±0.5 in C57BL/6J vs 2.1±0.4 in FVB/NJ (P>0.05) and 2.7±0.3 in FVB/NRj (p = 0.03); W20: 0.6±0.3 in C57BL/6J vs 1.9±0.4 in FVB/NJ (P>0.05) and 1.2±0.5 in FVB/NRj (p>0.05), and W24: 0.6±0.3 in C57BL/6J vs 2.4±0.3 in FVB/NJ (p = 0.0011 and 2.3±0.2 in FVB/NRj (p = 0.0051). Concerning their histological characterization, lesions appear as highly cellularized cushions both at the endothelial surface, which present a combination of flattened cells having the typical shape of endothelial cells and grouped round cells, and into a rich extracellular matrix where the cells are more star- or spindle-shaped (**Fig 3I and 3J**). Some cells are organized in extending spans in a proteoglycans rich matrix as well visualized by Alcian blue staining (**Fig 3G and 3H**). A link between leaflets thicknesses, platelet count and urinary 5-HIAA was found. When all data were pooled at 12, 16, 20 and 24 weeks a significant correlation was identified between thicknesses, urinary 5-HIAA (Spearman r = 0.2280, p = 0.0130) and platelet count (Spearman r = 0.2428, p = 0.0251 (**Fig 4**).

## Cardiac functional evaluation by echocardiography

To know if the mitral valve remodeling observed by histology was associated with functional cardiac alterations, echocardiography was performed in 10 C57BL/6J and 10 FVB/NJ, 24 week-old, male mice. This endpoint was selected according to our histological analysis. At that

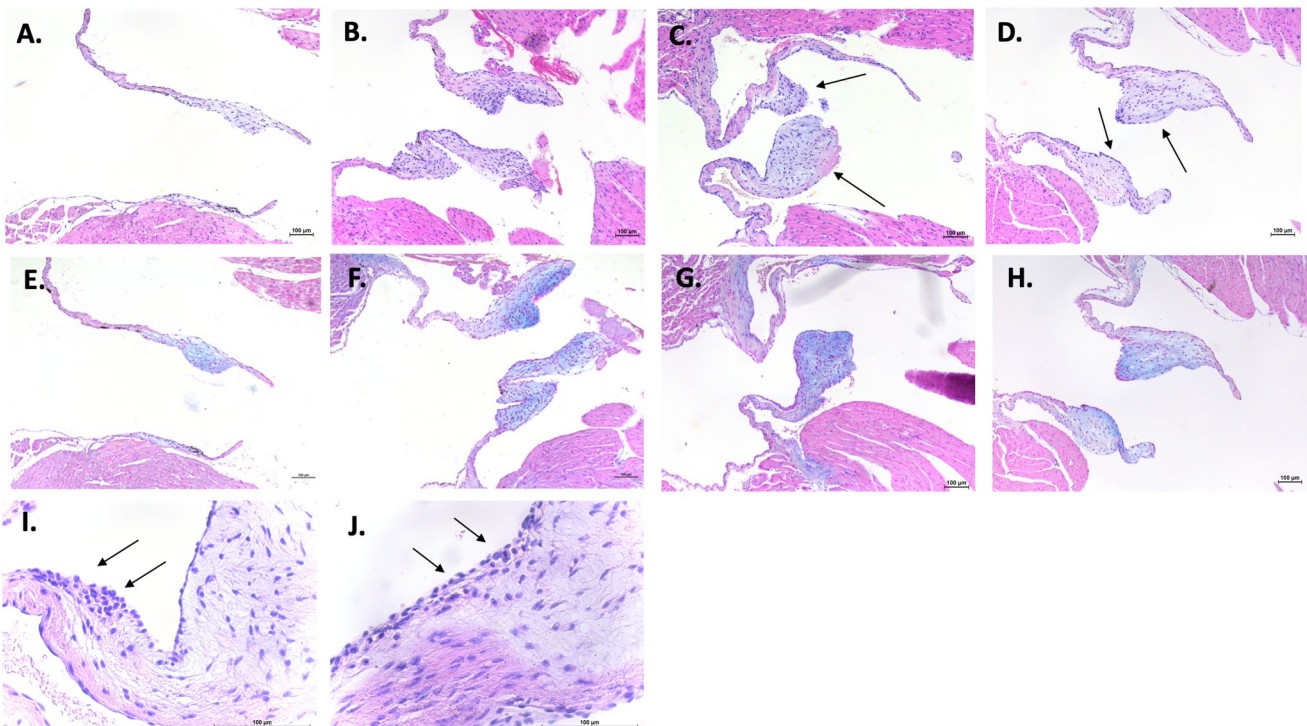

**Fig 3. Spontaneous mitral valve lesions in C57BL/6J and FVB mice.** Hematoxylin and eosin staining shows thickened leaflets, called "cushions", with important cellular density, in FVB/NJ (**C**) and FVB/NRj (**D**) mice, in comparison to thin leaflets in C57BL/6J mice (**A**). However, some C57BL/6J mice may have highly cellularized valvular lesions (**B**), but with a prevalence of only 30% and a lower thickness. In C57BL/6J (**E** and **F**), as well as in FVB mice (**G** and **H**), Alcian blue staining shows severe deposition of glycosaminoglycans (blue) in the spongiosa and disrupted, disorganized collagen (pink); all features that are characteristics of fibromyxoid lesions. Finally, clusters of rounded shape cells bordering the valvular surface and penetrating the interstitial matrix were observed in all mice with lesion, but more prominent in FVB mice (**I** and **J**) (scale bar = 100μm).

age, the two groups of animals showed a similar body weight (31.8±0.6g and 33.3±0.9g in C57BL/6J and FVB/NJ respectively, p>0.05). Compared to C57BL/6J, FVB/NJ animals demonstrated an end-diastolic dilatation of the left ventricle (4.23±0.07mm in FVB/NJ *vs* 3.95 ±0.11 in C57BL/6J; p = 0.04) associated to a dilatation of the left atrium as attested by the increase in left atrial diameter (2.24±0.09mm in FVB/NJ *vs* 1.77±0.08 in C57BL/6J; p = 0.001)

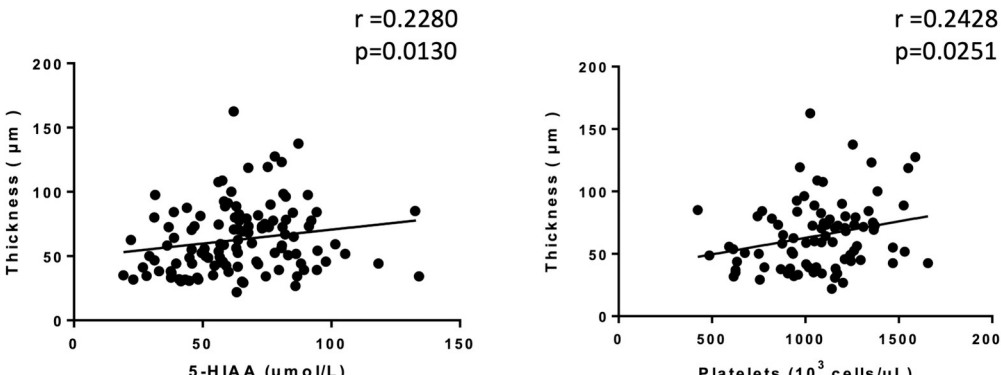

**Fig 4. Correlations between mitral valve thickness and either 5-HIAA or platelets.** Left: correlations between mitral valve thickness and pooled urinary 5-HIAA at 12-, 16-, 20- and 24- weeks (Spearman r = 0.2280, p = 0.0130). Right: correlation between thickness and pooled platelets count at 12-, 16- and 20- weeks (Spearman r = 0.2428, p = 0.0251).

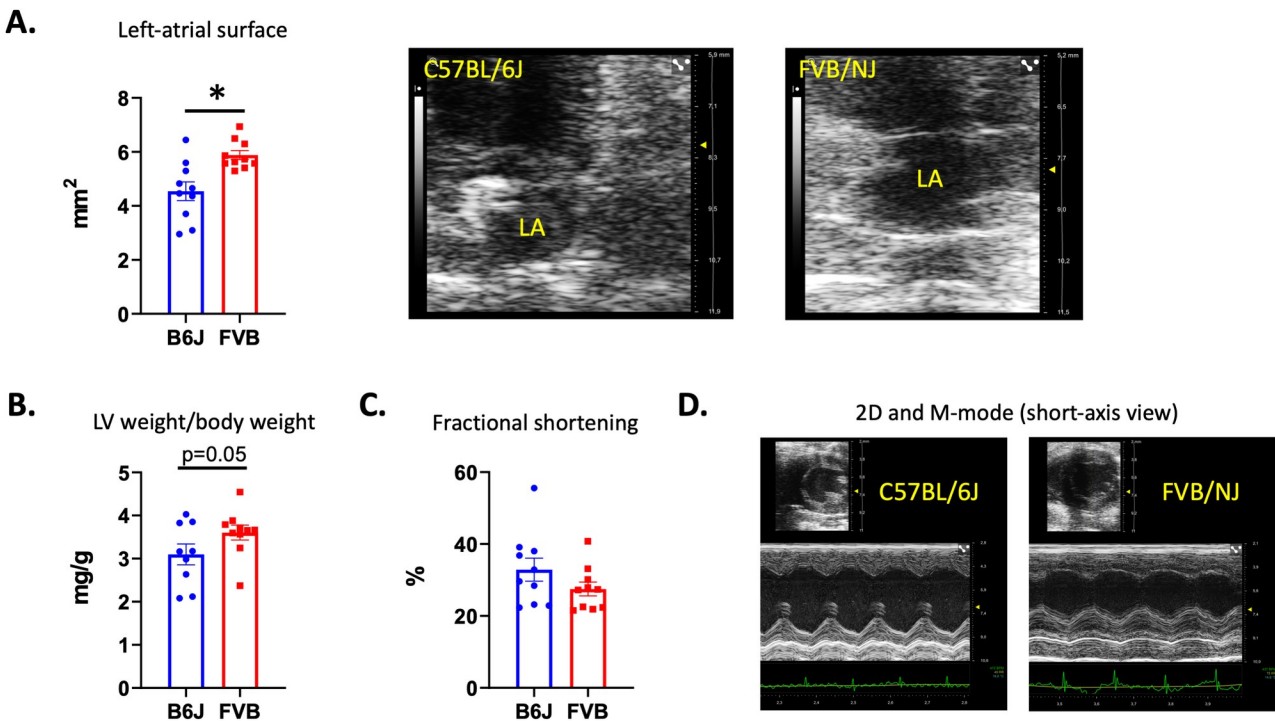

**Fig 5. Echocardiographic evaluation showing left cardiac cavities and systolic left ventricular function.** In FVB/NJ mice compared to C57BL/6J echocardiography identified the enlargement of the left atrium (A). Echocardiographic pictures are showing C57BL/6J (upper and left panel) and FVN/NJ (upper and right panel) left atrium obtained in a 2D apical view. An important dilation of the left-atrium is observed in the FVB/NJ mouse. We also observed a left-ventricular hypertrophy (B). At that stage, no reduction of the systolic function was observed (C). In D, the increased thickness of the ventricular walls can be observed on the 2D (upper picture) and the M-mode obtained in a mid-ventricular short axis section of the left ventricle in a C57BL/6J (left) and a FVB/NJ (right). A disorganization of papillary muscles can also be observed in the FVB/NJ mouse case.

and surface (**Fig 5A**). Moreover, we observed an increase in LVW (C57BL/6J and FVB/NJ showing respectively 101±7mg and 119±5mg, p = 0.04) and LVW on body weight ratio (**Fig 5B**) in FVB/NJ mice. The systolic function was preserved with no significant reduction of the SF (**Fig 5C and 5D**) and CO (24.39±2.01mL/min in FVB/NJ *vs* 30.52±2.82mL/min, in C57BL/6J; p>0.05).

At a functional point of view in these last animals, the speed of cardiomyocytes relaxation was reduced as attested by a significant increase of the IVRT (**Fig 6A and 6B**) which, was, at that stage, not associated with an increase in left ventricular filling pressures because the E/A ratio was unchanged (1.35±0.09 in FVB/NJ *vs* 1.38±0.08 in C57BL/6J; p>0.05). Noteworthy, during echocardiography, IVRT was measured in animals stabilized at a similar heart rate, respectively 424±14 bpm and 392±10 bpm in C57BL/6J and FVB/NJ (p = 0.09), a non-significant 7% difference that cannot by itself explain the IVRT increase. Nevertheless, the maximal speed of the aortic flow was diminished (641±16mm/s in FVB/NJ *vs* 848±70 in C57BL/6J; p = 0.004) attesting a reduction of the contractility of the cardiomyocytes probably mainly affecting longitudinal fibers because SF was preserved. Concerning the mitral valve apparatus, we observed a non-significant augmentation of the mitral annulus diameter (**Fig 6C**) and a marked thickening of both mitral valve leaflets (**Fig 6D**) in FVB/NJ mice. Together with the absence of these typical anatomic changes observed in a mitral valve prolapse, we confirmed the absence of such a prolapse by the observation of the lack of systolic atrial protrusion of neither the anterior nor the posterior mitral valve in FVB/NJ animals (**Fig 7A**). In our experimental series, 30% of male FVB/NJ mice were demonstrating mitral insufficiency as shown by

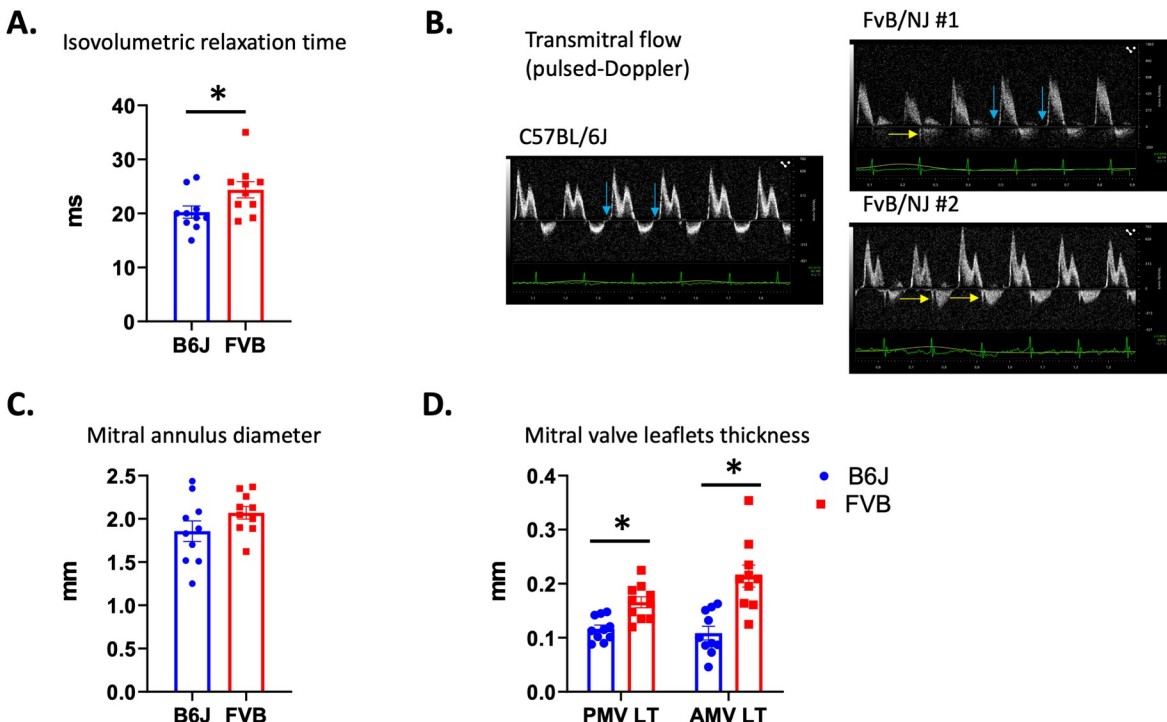

**Fig 6. Diastolic function and anatomo-pathologic parameters of the mitral apparatus.** A trouble of the cardiomyocytes primary relaxation as attested by the isovolumetric relaxation time increase (**A**) was observed in FVB/NJ animals. This increase is shown in the FVB/NJ case #1 (blue arrows) (**B**). In the two FVB/NJ mice a mitral insufficiency is detected with a pulsed-Doppler window placed at the tip of the leaflets meaning that the insufficiency is originating at the valve's commissure (yellow arrows). Mitral valve annulus is not enlarged (**C**) but leaflets are thickened (**D**).

color Doppler (**Fig 7B**) when none was observed in age-matched male C57BL/6J. To date, there is no classification of mitral insufficiency in mice and the grade could only be obtained by referring to the guidelines of the American Society of Echocardiography in humans. Three grades a usually employed. If the backflow is small, central and covers less than 20% of the atrial surface, the insufficiency is classified as Light. If the surface varies depending on loading conditions, the insufficiency is Moderate. Finally, in severe mitral insufficiency, the direction of the flow is central or lateral, covering more than 40% of the atrial surface. Fig 7B clearly shows a typical severe case and all our cases can be classified as Moderate or Severe. By plotting EDLVD and the thickness of the anterior mitral leaflet we found a highly significant correlation ($R^2$ = 0.33, p = 0.009) (**Fig 7C**). Conversely, no correlation was found between EDLVD and the diameter of the mitral annulus. Taken together, the echocardiographic data show that, compared to C57BL/6J mice, FVB/NJ demonstrate a marked thickening of the mitral valve leaflets that is associated in one third of the animals with mitral insufficiency. These alterations have hemodynamic consequences and cardiac remodeling with dilatation of both atrial and ventricular cavities. At that stage, only subtle alterations of cardiomyocyte function were observed in diastole (increase in IVRT) and systole (decrease of the peak aortic flow velocity).

## Discussion

In this work, we have characterized two FVB mouse substrains *i.e.* FVB/NJ and FVB/NRj in comparison with C57BL/6J, the usual background strain of numerous cardiovascular studies. We have shown that the FVB genetic background demonstrates a high prevalence of mitral valve lesions (80–100% of the animals). No significant gender difference was observed.

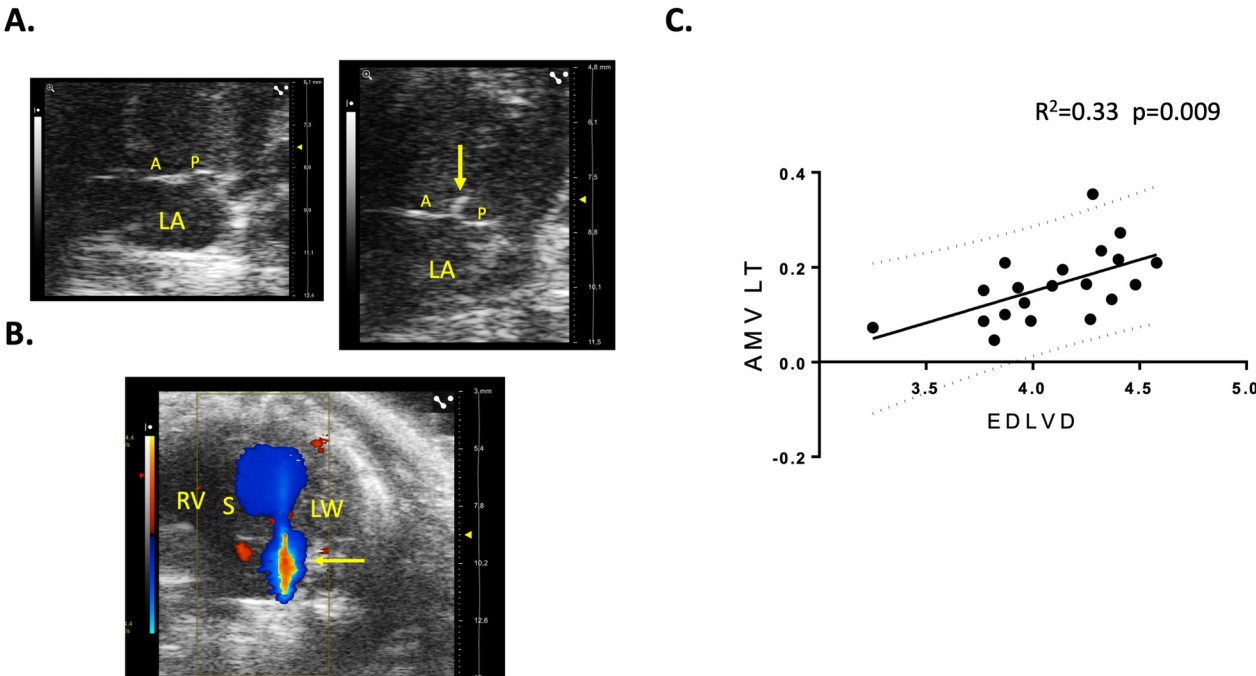

**Fig 7. Mitral valve insufficiency in FVB/NJ mice and correlation between left-ventricular dilatation and mitral valve thickening.** The absence of mitral valve prolapse is demonstrated by the lack of systolic valvular protusion towards the atrium in FVB/NJ (right) compared to C57BL/6J (left) (apical view). Note the mitral lesions observed at the tip of mitral leaflets (arrow) in the FVB/NJ case (**A**). Some FVB mice show massive mitral valve insufficiency as shown by the systolic backflow from the ventricle to the atrium observed with color Doppler (yellow arrow). The red part in the middle of the blue backflow corresponds to aliasing due to high flow velocities (apical view) (**B**). The remodeling of the mitral valve is associated with the dilatation of the left ventricle (**C**). A: anterior leaflet of the mitral valve, P: posterior leaflet of the mitral valve, LA: left atrium, S: interventricular septum, RV: right ventricle, LW: lateral wall of the left ventricle, AMVLT: Anterior Mitral Valve Leaflet Thickness, EDLVD: End-Diastolic Left-Ventricular Diameter.

Mitral valve disease is a complex entity with a variety of origins. Degeneration can occur following an extrinsic attack (infection, autoimmune involvement, medication), on an isolated malformation or associated with a polymalformative syndrome or due to a hemodynamic anomaly which disturbs its structure and/or function (left ventricular dilation, myocardial infarction) [10]. The most frequent degeneration is called fibromyxoid affecting 2.4 to 5% of the population, making the bed for acquired mitral leakage in the elderly [11]. This spontaneous degeneration appears as thickened, gelatinous lesions looking like pads applied at the valve surface. It can cause displacement of the anterior valve to the left atrium causing prolapse. In humans, this histological description is typical although not specific since it is found in other situations such as rheumatic valve disease. There is a disorganization of the collagen framework which is replaced by an intense overload of proteoglycans which itself makes the bed for fibrosis. The lesions that we observed in our study are very similar to this description since they are made by cellular pads where cells of fibroblastic appearance are distributed in a proteoglycan-rich matrix as seen by Alcian blue staining. The endothelial surface shows both typical endothelial cells and round cells in which we have identified an important CD34/CD31 labelling, indicating their endothelial progenitor lineage [9]. Similar lesions were also observed in dogs with the so-called degenerative myxomatous mitral valve disease where collagen matrix appears disorganized, replaced by glycosaminoglycans with proliferation of stromal and endothelial cells [12]. Similar to the dog's description, we have observed different stages of the lesions, the amount of glycosaminoglycans deposits being linked to the severity. FVB mice could therefore be a model of spontaneous myxomatous mitral valve degeneration without

mitral valve prolapse because no increase of leaflets length, no annular dilatation and no systolic atrial protrusion were identified.

The origin of this degeneration is therefore elusive. FVB mice were obtained from the breeding of the two ancestral strains HSFR/N and HSFS/N derived from a stock of Swiss mice. In the 8[th] generation, the presence of the Fv1b allele, which induces sensitivity to strain B of the Friend's leukemia virus, is demonstrated on the HSFS/N strain. Mice homozygous for this allele will then be inbreeded to create the FVB/N strain. Its main phenotypic characteristics include a respiratory system highly sensitive to asthma with a significant production of specific IgE. Finally, at a genetic point of view, the strain is homozygous for the Pde6brd1 allele which causes retinal degeneration. No other clear phenotype has yet been identified and, in particular, no link has never been established between the two known mutated genes and any cardiovascular alteration. A high IgE production was associated to rapidly progressing valve bioprosthesis degeneration in sheep [13] and hypereosinophily is a risk factor for mitral valve fibrosis [14]. In our work, we did not measure plasma IgE but none of the FVB animals exhibited hypereosinophily, the eosinophil blood count being even lower in FVB than C57BL/6J. We also did not identify a cardiovascular risk factor for valve disease such as hypertension. This work was not intended to explore other cardiovascular risk factors in these animals but a recent study was interested in the basic phenotype and following a fructose-enriched diet for 12 weeks in FVB/NJ mice compared to C57BL/6J [15]. This study showed that the FVB mice have a hypercholesterolemia carrying on both the HDL and LDL fractions of cholesterol. This increase could favor the development of valve lesions but, these mice have been shown to be resistant to atherosclerosis [16]. Furthermore, they have no abnormality in carbohydrate metabolism and overproduce Fgf21, a protective factor in the metabolic syndrome [17]. It is therefore unlikely that metabolic abnormalities are responsible for the valve phenotype that we describe.

The high amounts of proteoglycans in myxomatous lesions could direct towards a contribution of the serotonergic system. Indeed, valve remodeling observed in the myxoid valves is particularly rich in decorin, biglycan and versican. In addition, clearance of hyaluronan is reduced due to the decreased expression of the receptor involved in its endocytosis (HARE for hyaluronan receptor for endocytosis) [11]. The origin of this increase in proteoglycans and hyaluronan deposits is not fully understood but a contribution of the serotonergic system is suggested. Indeed, in dogs, the formation of myxoid mitral lesions has been associated with an increase in the concentration of serotonin in platelets and with a modification of the expression of key proteins of the serotonergic system in the valve tissue with increase in the 5-HT$_{2B}$ receptor and repression of the SERT transporter [5, 18, 19]. Abnormalities in serotonin clearance in the valve tissue could explain the increased serotonin concentration in the valve tissue itself [19]. Activation of the 5-HT$_{2B}$ receptor could lead to the production of proteoglycans as has been suggested in pig valves organotypic cultures [20] even if the effects were not spectacular in this latter work, probably due to the mediator's metabolism and the absence of mechanical constraints. Indeed, the latter seem fundamental for obtaining full induction of synthesis of extracellular matrix by serotonin [21]. In addition, several serotonergic receptors are involved (5-HT$_{2A}$ and 5-HT$_{2B}$) as well as circulating cells not mobilized in an *ex vivo* system [9]. Similarly, chronic stimulation of serotonergic 5-HT$_2$ receptors by drugs or recreational substances such as ecstasy [9, 22] or methamphetamine [23] cause myxoid mitral valve damage in humans. Serotonin itself can cause heart lesions including the 4 valves in the so-called carcinoid heart linked to tumors secreting large amounts of serotonin [24]. The cellular and molecular mechanisms linking serotonin and myxoid lesions are not yet fully understood, but TGF-β1 seems to be the cornerstone of the activation of cells of the extracellular matrix by this mediator in the valve [25] and the myocardium [26]. In the present work, we have shown that valve

lesions of FVB mice share all the characteristics of myxoid lesions and we have therefore hypothesized that the serotonergic system is involved in the initiation and development of remodeling. In phenotypic analysis studies, an activation of this system has already been shown in FVB/NJ mice compared to C57BL/6J. Thus, it has been observed that FVB/NJ mice show a more than 10x reduction of the immobility time in a forced swim test compared to C57BL/6J mice meaning that they are less depressed. Moreover, they do not respond to desipramine or fluoxetine, two antidepressants that inhibit the serotonin transporter [27]. Thus, these animals would spontaneously exhibit higher concentrations of extracellular free serotonin than C57BL/6J animals. The lack of pharmacological response to a blocker could indicate a decrease in serotonin transporter activity. In another regulation, this central serotonergic system appears to be well activated in this strain where it contributes to respiration, particularly in the immediate postnatal period. In FVB/NJ mice, neurons of the medulla oblongata involved in respiration show elevated cellular contents of serotonin compared to C57BL/6J mice [28]. A work in autism showed that the reduction in social interactions caused by partial Fmr1 protein deficiency occurs in a mixed FVB/N-129/OlaHsd background in a similar manner then in partial serotonin transporter (Slc6a4-/y) deficient mice and not in C57BL/6J animals [29]. This result is consistent with a deficit in SERT activity in FVB/NJ mice since this has never been described in 129/OlaHsd. In the periphery, reduction of serotonin transporter activity by pharmacological blockers or genetic invalidation results in a significant increase in bleeding time as we observed in the present work [30]. Thus, FVB/NJ mice would have normal or increased serotonin synthesis as evidenced by normal total serotonin concentrations in the blood of our animals. A possible partial deficit in serotonin transporter activity would not prevent platelets from serotonin loading, the overall increased platelet content coming from the thrombocytosis observed in these animals. This reuptake deficit could also explain the increased serotonin catabolism evidenced by the increase in urinary 5-HIAA in our work. This molecule is a product of the metabolism of serotonin whose stability in urines allows to have an appreciation of the level of activity of the serotonergic system as it is now clearly established in carcinoid heart disease [31]. We were able to show a link between the severity of the valve lesions, the platelet count and the urinary 5-HIAA thus showing the probable activation of the serotonergic system. This increased catabolism must be related to increased tissue uptake of serotonin in clearance organs such as the lung, possibly explaining the greater susceptibility of these mice to the development of hypoxic pulmonary hypertension than C57BL/6J [32]. In the valves, FVB/NJ mice could therefore also take up more serotonin, a mediator that could more easily engage its pharmacological targets such as the 5-HT$_{2A}$ and 5-HT$_{2B}$ receptors detected in this work within the anterior leaflet of the mitral valve but also exert deleterious intracellular effects by increasing oxidative stress or serotonylation of intracellular proteins already described in the development of fibromyxoid lesions of this valve [33]. Taken together, literature and our data argue in favor of a major dysregulation of the serotonergic system in FVB/NJ mice that could explain their higher sensitivity to spontaneous mitral valve degeneration. The tissue and cellular impact of this system in FVB mice is not clear and we will need to know if an increase in its effects contributes to more mobilization of endothelial progenitors from the bone marrow as we have seen in 129S2/SvPAS mice [9] or reinforces local effects within the valve tissue itself. Unexpectedly, we observed lesions in the C57BL/6J mice which constituted our genetic background control. We decided to choose this mouse strain in the light of a previous work which showed the absence of mitral valve thickening over time in these animals [34] and because they have very similar morphological and behavioral characteristics to FVB. Our work shows that they exhibit lesions smaller and less frequent than FVB, a situation that is nevertheless different from the 129S2/SvPAS control that we had previously used and in which we never observed any spontaneous lesion. We suspected that the

mitral valve lesions we observed in FVB induce hemodynamic consequences because a progressive pro-ANP increase was noticed. To confirm this hypothesis, we performed echocardiographic examinations in the most severely affected strain *i.e.* the FVB/NJ at 24 weeks of age to fit with our histological endpoint. Indeed, compared to C57BL/6J, we observed an important left-atrium enlargement that could by itself explain the pro-ANP increase. Moreover, the left ventricle was also slightly dilated and showed a concentric hypertrophy. At that stage left ventricular function was still preserved. We only observed a reduction of the speed of cardiomyocytes relaxation that is typical of ventricular hypertrophy. Interestingly, we noticed a correlation between ventricular dilatation and the thickening of mitral valve leaflets that could argue in favor of a secondary functional mitral insufficiency due the ventricular dilatation. In fact, we did not observe any dilatation of the mitral annulus confirming that the mitral valve thickening is neither due to the dilatation of the ventricle nor to a congenital mitral valve prolapse, the absence of annulus dilatation and systolic protrusion of mitral leaflet combined to the lack of increase in leaflets length ruling out the hypothesis of such a prolapse.

In conclusion, the FVB mouse strain is highly prone to spontaneous mitral valvulopathy, the substrain coming from the Jackson's laboratory (delivered by Charles River in France) being more affected. C57BL/6J mice used as a control is probably not the best one or at least is a control that is also showing lesions and remodeling with a lower prevalence and severity than FVB. For this aim, 129S2/SvPAS is probably a better choice [9]. The question of the interest of this new model to test new drugs dedicated to the prevention and/or treatment of mitral valve degeneration is now opened and will be investigated in further studies. The interest of the model to screen *in vivo* drugs at risk to increase preexisting lesions is also an opportunity of this model.

## Supporting information

**S1 Fig. Heart to body weight ratio from C57BL/6J and FVB mice at 12, 16, 20 and 24 weeks.** This ratio appreciates myocardial hypertrophy. No significant difference is observed among groups.
(TIF)

**S2 Fig. Evaluation of myocardial collagen content between C57BL/6J and FVB mice at 12, 16, 20 and 24 weeks by picro-Sirius red staining.** Results are expressed as a percentage of collagen par surface unit. Following cardiac growth, the percentage of collagen decreases because of the physiological cardiac hypertrophy. Nevertheless, no difference was observed between groups at any given time point between 12 and 24 weeks.
(TIF)

## Author Contributions

**Conceptualization:** Estelle Ayme-Dietrich, Guillaume Becker, Laurent Monassier.

**Data curation:** Estelle Ayme-Dietrich.

**Formal analysis:** Estelle Ayme-Dietrich, Laurent Monassier.

**Funding acquisition:** Laurent Monassier.

**Investigation:** Sylvia Da Silva, Ghina Alame Bouabout, Alizée Arnoux, Laurent Monassier.

**Methodology:** Laurent Monassier.

**Supervision:** Laurent Monassier.

**Validation:** Laurent Monassier.

**Writing – original draft:** Estelle Ayme-Dietrich, Laurent Monassier.

**Writing – review & editing:** Estelle Ayme-Dietrich, Jérôme Guyonnet, Guillaume Becker.

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
