## [Decision Letter · Decision Letter 0]

1 Jun 2021

PONE-D-21-13499

Characterization of the spontaneous degenerative mitral valve disease in FVB mice: a new model for pathophysiological and pharmacological studies

PLOS ONE

Dear Dr. Monassier,

Thank you for submitting your manuscript to PLOS ONE. After careful consideration, we feel that it has merit but does not fully meet PLOS ONE’s publication criteria as it currently stands. Therefore, we invite you to submit a revised version of the manuscript that addresses the points raised during the review process.

All reviewer's comments need to be addressed.

We look forward to receiving your revised manuscript.

Kind regards,

Cécile Oury

Academic Editor

PLOS ONE

Journal Requirements:

2. Thank you for including your ethics statement:  "All animal care and procedures are in accordance with institutional guidelines and European regulations. The protocol has been submitted to French regulation authorities and ethical committee (CREMEAS) according to the new European guidelines. It obtained its authorization from the Ministère de l’Enseignement Supérieur, de la Recherche et de l’Innovation with the following registration number 11732-2017121309379999.".   

Please amend your current ethics statement to confirm that your named ethics committee specifically approved this study.

For additional information about PLOS ONE submissions requirements for ethics oversight of animal work, please refer to http://journals.plos.org/plosone/s/submission-guidelines#loc-animal-research  

3.Thank you for stating the following in the Acknowledgments Section of your manuscript:

"They also thank CEVA Santé

456 Animale (Libourne, France), the Fédération Française de Cardiologie (FFC) and the Medical Faculty

457 and the University of Strasbourg for financial and administrative support."

 "The authors received no specific funding for this study"

Reviewers' comments:

Reviewer's Responses to Questions

**Comments to the Author**

1. Is the manuscript technically sound, and do the data support the conclusions?

Reviewer #1: Partly

2. Has the statistical analysis been performed appropriately and rigorously? 

Reviewer #1: No

3. Have the authors made all data underlying the findings in their manuscript fully available?

Reviewer #1: No

4. Is the manuscript presented in an intelligible fashion and written in standard English?

Reviewer #1: Yes

5. Review Comments to the Author

Reviewer #1: This is an observational animal study conducted at Strasbourg University in France to investigate mitral valve degeneration in FVB strains and C57BL/6J strain mice. Blood and urine samples, echocardiographic data, and histological analyses were compared between the strains at 12-24 weeks. The authors report left ventricular (LV) remodeling, increased LV weight, and increased mitral valve thickness in the FVB group compared to the C57BL/67 mice, together with the presence of mitral regurgitation (MR) in 30% of FVB mice. In addition, they reported increased 5-HIAA and platelet counts in the FVB mice, suggesting a possible contribution of systemic serotonergic activation.

Surgical treatment, including endovascular treatment, is the gold standard for degenerative MR. However, when degeneration progresses to bileaflet lesions, repair surgery becomes more challenging. Therefore, the prognosis of patients with MR may be improved if degeneration can be prevented or treated with drugs. The creation of an experimental animal model for this purpose is interesting from a translational perspective. Therefore, the manuscript is studying an important topic. Unfortunately, although it seems like the FBV strains develop increased mitral valve remodeling, the presented data was not very strong and there were inappropriate data interpretations throughout the manuscript. In humans, serotonin is inactivated in the lungs and mostly affects the tricuspid valve, suggesting that myxomatous changes in the mitral valve are generally attributed to reasons other than serotonin. More strong and direct evidence of serotonin involvement needs to be presented to support its contribution.

The following are critiques/suggestions for improving the Manuscript.

Major

1. Data interpretation is inappropriate in many places:

- Line 208 “tendency to a higher concentration being observed in FVB animals”. There is a large variability in 5-HIAA level for each time point and there seems to exist no consistent tendency that indicates clear upregulation of 5-HIAA only in the FVB lines. Line 209 “This increase reached statistical significance in pooled FVB/NJ vs C57BL/6J at W20” is likely by chance, since the C57BL/6J animals also show that level of 5-HIAA at other time points. (it is also not corrected for multiple comparisons). Same applies to pro-ANP.

-Line 217 “their platelets release more 5-HT and/or have a reduced 5-HT loading capacity” is not supported by data.

2. Urine 5-HIAA is an indirect method and does not necessary indicate that it contributed to the mitral valve remodeling. Acutual serotoning signaling in mitral valve needs to be examined for this purpose.

3. Line 304 “we observed an increase in LVW on body weight ratio (Figure 4B) in FVB/NJ mice”. In the absence of difference in actual LV weight, this only suggests the bias in echo results.

4. Mitral leaflet thickness is already increased at 12 weeks in FVB lines and no clear separation between the strains can be found thereafter from the Figure. This questions if the model is suitable for testing therapeutic approaches to prevent mitral valve remodeling, at least within the studied age range. (Authors state 91.8% increase at week 24, but the average thickness in FBV/NJ is not different from week12.)

Minor

5. The presented data does not support the utility of FBV mice as a model to study pharmacological studies. Please remove the subtitle.

6. There are several paragraphs with "data not shown". Authors should make efforts to include them in the supplemental data.

7. Please describe actual p values instead of stating larger or smaller than 0.05.

8. Fig. 1B is mentioned after Fig.3 in the manuscript. Since Fig 1B is not so related to Fig 1A, please move this fig in the order it appears.

9. Please clarify the grade of MR.

10. Line 191 “part of the HR increase was compensated by a blood pressure reduction” is not an appropriate statement which is unlikely to be true and ignores the complex cardiovascular physiology.

11. Line 251, “Therefore, this result shows the absence of mitral valve prolapse in FVB compared to our C57BL/6J control”. To begin with, "prolapse" is a word to describe leaflet motion. How did the author judge the presence of prolapse with histopathological assessment?

12. Fig3 control Alcian blue staining should be presented. The legend statement “Finally, clusters of rounded shape cells bordering the valvular surface and penetrating the interstitial matrix were observed in FVB mice” is probably not specific to FVB mice and control seems to also have the clusters.

13. Fig 4. Actual echo images should be accompanied to support the statements. Please include the labels for what is measured in each graph.

14. IVRT is unlikely to be reliable in such high HR conditions with difference in HR between the groups

6. PLOS authors have the option to publish the peer review history of their article (what does this mean?). If published, this will include your full peer review and any attached files.

Reviewer #1: No

---

## [Author Response · Author response to Decision Letter 0]

18 Aug 2021

The authors would like to thank the referee for all the work performed and for the contribution to the improvement of our article. We have performed additional experiments and deeply modified our text. All changes appear in red in the corrected version of our paper.

Reviewer #1: This is an observational animal study conducted at Strasbourg University in France to investigate mitral valve degeneration in FVB strains and C57BL/6J strain mice. Blood and urine samples, echocardiographic data, and histological analyses were compared between the strains at 12-24 weeks. The authors report left ventricular (LV) remodeling, increased LV weight, and increased mitral valve thickness in the FVB group compared to the C57BL/67 mice, together with the presence of mitral regurgitation (MR) in 30% of FVB mice. In addition, they reported increased 5-HIAA and platelet counts in the FVB mice, suggesting a possible contribution of systemic serotonergic activation. Surgical treatment, including endovascular treatment, is the gold standard for degenerative MR. However, when degeneration progresses to bileaflet lesions, repair surgery becomes more challenging. Therefore, the prognosis of patients with MR may be improved if degeneration can be prevented or treated with drugs. The creation of an experimental animal model for this purpose is interesting from a translational perspective. Therefore, the manuscript is studying an important topic. Unfortunately, although it seems like the FBV strains develop increased mitral valve remodeling, the presented data was not very strong and there were inappropriate data interpretations throughout the manuscript. In humans, serotonin is inactivated in the lungs and mostly affects the tricuspid valve, suggesting that myxomatous changes in the mitral valve are generally attributed to reasons other than serotonin. More strong and direct evidence of serotonin involvement needs to be presented to support its contribution.

The following are critiques/suggestions for improving the Manuscript.

Major

1. Data interpretation is inappropriate in many places:

- Line 208 “tendency to a higher concentration being observed in FVB animals”. There is a large variability in 5-HIAA level for each time point and there seems to exist no consistent tendency that indicates clear upregulation of 5-HIAA only in the FVB lines. Line 209 “This increase reached statistical significance in pooled FVB/NJ vs C57BL/6J at W20” is likely by chance, since the C57BL/6J animals also show that level of 5-HIAA at other time points. (it is also not corrected for multiple comparisons). Same applies to pro-ANP.

Authors reply: We thank the reviewer for this comment and would like to point out that ANOVA tests are corrected for multiple comparisons with Dunn’s test, as stated in Methods. However, we agree that the 5-HIAA dosages are not a sufficient argument to demonstrate the involvement of the serotonergic system. For this reason, additional experiments were performed and their results added in the paper (please see the following questions).

-Line 217 “their platelets release more 5-HT and/or have a reduced 5-HT loading capacity” is not supported by data.

Authors reply: We fully agree with this referee’s comment. Urinary 5-HIAA, that is a reliable marker of global serotonergic system activity, and blood platelet were only presented in this first version of our paper. To answer this comment, we decided to more deeply investigate the serotonergic system in these animals and made the following additional experiments:

• Measurement of the bleeding time to assess platelet activation and global function

• RT-ddPCR quantification of mRNA encoding 5-HT2A and 5-HT2B receptors in the anterior leaflet of the mitral valve

• Measurement of whole blood and plasma serotonin concentrations allowing the determination of global platelet serotonin content

Therefore, our article now includes:

• The detailed experimental methods describing the additional techniques for droplet RT-ddPCR, bleeding time and the measurement of serotonin concentration in the blood and plasma. The in vivo methods are also completed for the mitral valve sampling technique.

• The results include the 3 additional experiments added and are expressed as text

• A new paragraph now appears in the discussion. It argues on the role of the serotonergic system in the valvulopathy exhibited by these animals on the basis of added literature data and the new data collected to answer the referee’s comment

To summarize our results, we have observed a huge increase of the bleeding time in FVB mice compared to C57BL/6 (+60%). This increase is associated with a massive reduction of plasma free serotonin without any change of the whole blood serotonin concentration meaning that the turnover of 5-HT between platelets and plasma is markedly altered in FVB animals. Finally, at the mRNA level, we did not observe any difference regarding the expression of the 5-HT2A and 5-HT2B receptors in the anterior leaflet of the mitral valve. As described in other species, we show for the first time in mice that the expression of the 2A subtype is higher than the 2B one.

The new paragraph inserted in the discussion is the following. It contains additional references that were inserted in the bibliography. We also completed the abstract section with the new investigations and results.

“In phenotypic analysis studies, an activation of this system has already been shown in FVB/NJ mice compared to C57BL/6J. Thus, it has been observed that FVB/NJ mice show a more than 10x reduction of the immobility time in a forced swim test compared to C57BL/6J mice meaning that they are less depressed. Moreover, they do not respond to desipramine or fluoxetine, two antidepressants that inhibit the serotonin transporter (Lucki et al. 2001). Thus, these animals would spontaneously exhibit higher concentrations of extracellular free serotonin than C57BL/6J animals. The lack of pharmacological response to a blocker could indicate a decrease in serotonin transporter activity. In another regulation, this central serotonergic system appears to be well activated in this strain where it contributes to respiration, particularly in the immediate postnatal period. In FVB/NJ mice, neurons of the medulla oblongata involved in respiration show elevated cellular contents of serotonin compared to C57BL/6J mice (Menuet et al., 2011). A work in autism showed that the reduction in social interactions caused by partial Fmr1 protein deficiency occurs in a mixed FVB/N-129/OlaHsd background in a similar manner then in partial serotonin transporter (Slc6a4-/y) deficient mice and not in C57BL/6J animals (Moy et al, 2009). This result is consistent with a deficit in SERT activity in FVB/NJ mice since this has never been described in 129/OlaHsd. In the periphery, reduction of serotonin transporter activity by pharmacological blockers or genetic invalidation results in a significant increase in bleeding time as we observed in the present work (Oliver et al, 2016). Thus, FVB/NJ mice would have normal or increased serotonin synthesis as evidenced by normal total serotonin concentrations in the blood of our animals. A possible partial deficit in serotonin transporter activity would not prevent platelets from serotonin loading, the overall increased platelet content coming from the thrombocytosis observed in these animals. This reuptake deficit could also explain the increased serotonin catabolism evidenced by the increase in urinary 5-HIAA in our work. This molecule is a product of the metabolism of serotonin whose stability in urines allows to have an appreciation of the level of activity of the serotonergic system as it is now clearly established in carcinoid heart disease 24. We were able to show a link between the severity of the valve lesions, the platelet count and the urinary 5-HIAA thus showing the probable activation of the serotonergic system. This increased catabolism must be related to increased tissue uptake of serotonin in clearance organs such as the lung, possibly explaining the greater susceptibility of these mice to the development of hypoxic pulmonary hypertension than C57BL/6J (Tada et al, 2008). In the valves, FVB/NJ mice could therefore also take up more serotonin, a mediator that could more easily engage its pharmacological targets such as the 5-HT2A and 5-HT2B receptors detected in this work within the anterior leaflet of the mitral valve but also exert deleterious intracellular effects by increasing oxidative stress or serotonylation of intracellular proteins already described in the development of fibromyxoid lesions of this valve (Sauls et al, 2012). Taken together, literature and our data argue in favor of a major dysregulation of the serotonergic system in FVB/NJ mice that could explain their higher sensitivity to spontaneous mitral valve degeneration.”

2. Urine 5-HIAA is an indirect method and does not necessary indicate that it contributed to the mitral valve remodeling. Actual serotonin signaling in mitral valve needs to be examined for this purpose.

Authors reply: We thank the reviewer for this comment and agree that the 5-HIAA dosages are not a sufficient argument to demonstrate the involvement of the serotonergic system in mitral valve remodeling. For this reason, plasma and platelet serotonin assays were performed and added to the article.

Concerning the 5-HT2A and 5-HT2B receptor signaling (both receptors being involved in mitral degeneration) we did not observe a change in mRNA expression when reported to Hprt. This means that no cell increased its 5-HT2A and 5-HT2B receptor expression but does not rule out the target engagement and/or any 5-HT contribution to intracellular processes involving oxidative stress or protein serotonylation. This appears in the new paragraph added to the discussion section (see upper).

3. Line 304 “we observed an increase in LVW on body weight ratio (Figure 4B) in FVB/NJ mice”. In the absence of difference in actual LV weight, this only suggests the bias in echo results.

Authors reply: many thanks for this comment coming from the absence of LV mass results in our initial text. We now have included LVW in the results showing a significant difference between C57Bl6 and FVB respectively 101+/-7mg and 119+/-5mg, P=0.04. The following sentence now appears in the results section just before the results concerning LVW on body weight ratio in the echocardiography section.

Moreover, in the histological part of our results, the whole weight heart is indicated but not the LVW that is obtained by echocardiography.

4. Mitral leaflet thickness is already increased at 12 weeks in FVB lines and no clear separation between the strains can be found thereafter from the Figure. This questions if the model is suitable for testing therapeutic approaches to prevent mitral valve remodeling, at least within the studied age range. (Authors state 91.8% increase at week 24, but the average thickness in FBV/NJ is not different from week12.)

Authors reply: As suggested by the referee, FVB mice already exhibit mitral lesions at 12weeks. We understand that the comment related to the separation of the two strains is related to the comparison of the two FVB strains. For sure, all FVB animals develop mitral lesions and we were unable to separate them. Nevertheless, the most important fact between 12 and 24 weeks is the incidence of the lesions because all FVB animals were affected at the age of 24 weeks and so this incidence was increasing over time. Concerning the question of the suitability of the model in testing drugs, we believe that two aspects should be considered. If we want to test a prevention approach, it is quite clear that drugs should be started earlier than 12 weeks. At the opposite, a curative strategy could be employed to test if and how drugs could reverse the myxoid part of fibromyxoid lesions, the fibrotic part being probably highly challenging to reverse.

We simplified our text and left the question of the suitability of this model to test drug fully opened in the discussion. In the abstract, this opening sentence is withdrawn.

Minor

5. The presented data does not support the utility of FBV mice as a model to study pharmacological studies. Please remove the subtitle.

Authors reply: according with the referee’s request the subtitle is now withdrawn from the title of our article.

6. There are several paragraphs with "data not shown". Authors should make efforts to include them in the supplemental data.

Authors reply: we understand that the referee wants to see all data discussed and, as suggested, we added two supplemental figures and two lines of the table 1: Normalized heart weight are presented in Supplementary Figure S1, collagen results in Supplementary Figure S2. Hematocrit and hemoglobin dosages now appear in Table 1.

7. Please describe actual p values instead of stating larger or smaller than 0.05.

Authors reply: We thank the reviewer for this comment. The p values were described now in the article. 

8. Fig. 1B is mentioned after Fig.3 in the manuscript. Since Fig 1B is not so related to Fig 1A, please move this fig in the order it appears.

Authors reply: to follow this referee’s suggestion, we reordered the figures and Figure 1B is now Figure 4, the following numbers were modified accordingly, as well as the legends.

9. Please clarify the grade of MR.

Authors reply: color Doppler was used to detect mitral regurgitation in the apical four cavities view. When detected, the Nyquist limit was set in a way to see the maximal velocity and surface of the backflow. Taking the limitations of this method into account, we determined the insufficiency as present or not. To date, there is no classification of mitral insufficiency in mice and the grade could only be obtained by referring to the guidelines of the American Society of Echocardiography. Three grades are usually employed. If the backflow is small, central and covers less than 20% of the atrial surface, the insufficiency is classified as Light. If the surface varies depending on loading conditions, the insufficiency is Moderate. Finally, in severe mitral insufficiency, the direction of the flow is central or lateral, covering more than 40% of the atrial surface. The Figure 7B clearly shows a typical severe case and all our cases can be classified as Moderate or Severe. This precision now appears in the results section of our text: “To date, there is no classification of mitral insufficiency in mice and the grade could only be obtained by referring to the guidelines of the American Society of Echocardiography in humans. Three grades a usually employed. If the backflow is small, central and covers less than 20% of the atrial surface, the insufficiency is classified as Light. If the surface varies depending on loading conditions, the insufficiency is Moderate. Finally, in severe mitral insufficiency, the direction of the flow is central or lateral, covering more than 40% of the atrial surface. The Figure 7B clearly shows a typical severe case and all our cases can be classified as Moderate or Severe.”

10. Line 191 “part of the HR increase was compensated by a blood pressure reduction” is not an appropriate statement which is unlikely to be true and ignores the complex cardiovascular physiology.

Authors reply: we understand that our sentence was unclear. This comment was, of course, only related to the result obtained on the rate x pressure product where the increased heart rate is compensated (in the calculation) by the diminished blood pressure making a myocardial oxygen consumption similar between C57BL/6 and FVB. For sure, it was not a physiological demonstration. To clarify, the sentence was modified accordingly: “No difference was observed between C57BL/6J and FVB mice meaning that myocardial oxygen demand is similar between the two strains.”

11. Line 251, “Therefore, this result shows the absence of mitral valve prolapse in FVB compared to our C57BL/6J control”. To begin with, "prolapse" is a word to describe leaflet motion. How did the author judge the presence of prolapse with histopathological assessment?

Authors reply: We fully agree with this referee’s comment concerning the definition of a mitral valve prolapse that is ordinarily obtained by echocardiography. Nevertheless, some typical features such as leaflets lengthening and mitral annulus dilatation can be approached with histological techniques. In the classical definition, fibromyxoid degeneration of the leaflets enters into the definition of the mitral valve prolapse. Based on the remodeling and the absence of anatomic changes of leaflets length and mitral annulus dilatation, we assumed the absence of mitral valve prolapse. Of course, mitral valve prolapse was also studied using echocardiography. We did not observe leaflets lengthening and mitral annulus dilatation and, more importantly, the absence of systolic protrusion of neither the anterior or posterior valves in the atrium. Figure 7 is now completed with two 2D pictures of the mitral valve obtained in an apical view and closed in systole in a C57BL/6J and a FVB showing the lack of such a systolic protrusion. The following sentence now appears in the results section: “Together with the absence of these typical anatomic changes observed in a mitral valve prolapse, we confirmed the absence of such a prolapse by the observation of the lack of systolic atrial protrusion of neither the anterior nor the posterior mitral valve in FVB/NJ animals (Figure 7A).” Figure 7 and its legend were modified accordingly. Note that lesions can be observed at the tip of mitral leaflets in the FVB case (arrow).

12. Fig3 control Alcian blue staining should be presented. The legend statement “Finally, clusters of rounded shape cells bordering the valvular surface and penetrating the interstitial matrix were observed in FVB mice” is probably not specific to FVB mice and control seems to also have the clusters.

Authors reply: All images have been performed with Alcian Blue staining and according to the referee's request, those corresponding to B6 have been added to Figure 3. The legend statement was nuanced as follows: 

“Finally, clusters of rounded shape cells bordering the valvular surface and penetrating the interstitial matrix were observed in all mice with lesions, but more prominent in FVB mice.”

13. Fig 4. Actual echo images should be accompanied to support the statements. Please include the labels for what is measured in each graph.

Authors reply: We agree with the referee’s comment, actual echo images support graphs and were added to Figures 5 and 7. To better insert these images, we also built the new Figure 6.

14. IVRT is unlikely to be reliable in such high HR conditions with difference in HR between the groups 

Authors reply: we (Marzak et al., J of Hypertension, 2014 & 2015) and others (Martinez et al., J Appl Physiol, 1985; Ferreira Curry et al., J Am Soc Echocardiogr, 2005; Polegato et al., Cell Physiol Biochem, 2015; Szokol et al., Molecules, 2017; Litwin et al., J Am Coll Cardiol, 1996) have validated IVRT to measure the cardiomyocyte relaxation by echocardiography in hypertension or myocardial infarction in rats that also exhibit high heart rate. The same was observed in mice were IVRT is commonly employed to follow the diastolic function (Cole et al., J Mol Cell Cardiol, 2020; Corrigan et al., Reprod Sci, 2010; Regan et al., Am J Physiol Heart Circ Physiol, 2015). Moreover, IVRT is used as part of a myocardial performance index in mice (Zang et al., Am J Physiol Heart Circ Physiol, 2007). Noteworthy, in this work echocardiography is performed in anaesthetized animals only. This method allows the measurement of echocardiographic parameters in animals stabilized in a similar range for their heart rate. During echocardiography, in C57BL/6J and FVB/NJ the heart rate was respectively 424+/-14 bpm and 392+/-10bpm (P=0.09), a non-significant 7% reduction that cannot by itself explain the observed IVRT increase. Therefore, we understood that the heart rate data obtained during echocardiography were missing from our first version. To correct this problem, the following sentence now appears in the results section: “Noteworthy, during echocardiography, IVRT was measured in animals stabilized at a similar heart rate, respectively 424±14 bpm and 392±10bpm in C57BL/6J and FVB/NJ (P=0.09), a non-significant 7% difference that cannot by itself explain the IVRT increase.” In the new Figure 6 a typical example of such an increase is shown.

---

## [Editor Report · Decision Letter 1]

23 Aug 2021

Characterization of the spontaneous degenerative mitral valve disease in FVB mice

PONE-D-21-13499R1

Dear Dr. Monassier,

We’re pleased to inform you that your manuscript has been judged scientifically suitable for publication and will be formally accepted for publication once it meets all outstanding technical requirements.

Kind regards,

Cécile Oury

Academic Editor

PLOS ONE
---

## [Editor Report · Acceptance letter]

25 Aug 2021

PONE-D-21-13499R1 

Characterization of the spontaneous degenerative mitral valve disease in FVB mice 

Dear Dr. Monassier:

I'm pleased to inform you that your manuscript has been deemed suitable for publication in PLOS ONE. Congratulations! Your manuscript is now with our production department. 

Kind regards, 

on behalf of

Dr. Cécile Oury 

Academic Editor

PLOS ONE